# Phase Separation as a Driver of Stem Cell Organization and Function during Development

**DOI:** 10.3390/jdb11040045

**Published:** 2023-12-12

**Authors:** Amalia S. Parra, Christopher A. Johnston

**Affiliations:** Department of Biology, University of New Mexico, Albuquerque, NM 87131, USA; sancham9@unm.edu

**Keywords:** phase separation, stem cell, chromatin, cell fate, cell polarity, spindle orientation, asymmetric cell division

## Abstract

A properly organized subcellular composition is essential to cell function. The canonical organizing principle within eukaryotic cells involves membrane-bound organelles; yet, such structures do not fully explain cellular complexity. Furthermore, discrete non-membrane-bound structures have been known for over a century. Liquid–liquid phase separation (LLPS) has emerged as a ubiquitous mode of cellular organization without the need for formal lipid membranes, with an ever-expanding and diverse list of cellular functions that appear to be regulated by this process. In comparison to traditional organelles, LLPS can occur across wider spatial and temporal scales and involves more distinct protein and RNA complexes. In this review, we discuss the impacts of LLPS on the organization of stem cells and their function during development. Specifically, the roles of LLPS in developmental signaling pathways, chromatin organization, and gene expression will be detailed, as well as its impacts on essential processes of asymmetric cell division. We will also discuss how the dynamic and regulated nature of LLPS may afford stem cells an adaptable mode of organization throughout the developmental time to control cell fate. Finally, we will discuss how aberrant LLPS in these processes may contribute to developmental defects and disease.

## 1. Introduction

Membrane-bound cytoplasmic organelles are a defining feature of eukaryotic cell structure and organization. These distinct structures house unique complements of protein complexes necessary for specific biochemical reactions, thereby allowing for a division of labor that enhances the efficiency of many essential cellular functions. However, the number of organelles almost certainly underestimates the complexity of cellular organization. Indeed, how cytoplasmic processes that occur outside the confines of organelles might be afforded a similar level of organization as those within them is not explained by the organelles alone. Furthermore, the organelles themselves can have an internal organization unexplained by their defining membrane. For example, subnuclear organization has long been known to be highly complex and occurs in the absence of additional membrane structures within the boundary of the nuclear envelope itself. Such observations suggest additional organizational principles beyond the traditional organellar model.

A recent advancement in our appreciation of subcellular organization has come from the discovery of liquid–liquid phase separation (LLPS) as a mode of protein and nucleic acid behavior within cells, as well as under isolated in vitro conditions. Driven by specific and often dynamic molecular forces rather than by the formation of formal membranes, this process generates biomolecular condensates that allow for the emergence of discretely organized and localized protein and/or nucleic acid complexes both within and outside organelles (Figure 1). Advances in the LLPS field have shown that the specific physical nature of these processes can differ; thus, herein we will prefer to use the term ‘condensates’ as a broad term to describe phase separation processes. This review will describe the general features of condensates and the cellular processes they impact, while also highlighting examples within stem cells and their roles in development. We will discuss the role of condensates in developmentally relevant signaling pathways, chromatin organization and cell fate, and asymmetric cell division, as well as how they may contribute to cancer stem cell behaviors and disease.

In the following sections, we provide a discussion of key features of biomolecular condensates and important cellular processes they regulate, with highlights of examples from stem cell function during development.

## 2. Key Principles of Phase Separation in Cellular Organization

The organization of biochemical reactions outside the physical boundaries of cellular organelles, as well as the specialized organization of components localized within certain organelle membrane walls, are necessary for efficient output. This in turn ensures the proper control of key biological processes ranging from gene expression to cell signaling. The existence of condensates (or coacervates) has long been postulated as a mechanism for such segregation of cellular components. The past decade saw biomolecular condensation emerge as a biophysical model for an expanding list of cellular functions [1,2]. Condensate formation involves the physical demixing of specific components from others in a previously single, mixed phase. The result is a multi-phase solution with condensates existing in a physically distinct phase separate from those components that remain in the bulk solution (e.g., cytoplasm). The ‘liquid–liquid’ term in LLPS encapsulates the physical features of many, although not all, biological condensates. Such condensates exhibit properties of matter in a liquid state. For example, condensate components can typically easily rearrange, both within droplets as well as by exchanging components with the surrounding bulk environment. Also, droplets can deform in shape and often undergo fusion with one another (Figure 1). These processes, as well as the initial condensate formation itself, are driven by favorable free energy changes when demixed components are energetically stabilized (e.g., like in the classical ‘oil and vinegar’ scenario). A more in-depth discussion of the biophysical nature of condensates and these dynamic processes can be found in other recent reviews [3,4].

Studies have also detailed common molecular properties of condensates (Figure 1). One of the first classes of components to be widely appreciated in condensate biology is that of ribonucleoprotein (RNP) complexes. In this case, the addition of a cognate RNA often triggers the phase separation of its RNA-binding protein (RBP), with condensate size and number changing as a function of RNA concentration [5]. The electrical charge properties of the RNA confer significant changes to the RBP once bound, leading to energetically stable condensates consisting of RNP complexes. Another property common to RNP condensates is the presence of intrinsically disordered regions (IDRs) within the RBP protein sequence. Sequence analysis of RBP families revealed IDRs as a common feature, and the removal of disordered regions typically impaired condensate formation [6]. Unique sequence codes within these IDRs have also been identified, with specific glycine, serine, glutamine, arginine, and tyrosine motifs being common [7,8,9]. Thus, specific molecular aspects of both protein and RNA sequences influence the phase separation of RNP complexes.

Stem cells express a diverse repertoire of RBPs that perform critical functions, including in the regulation of stemness and differentiation [10,11,12]. Evidence for condensate formation has been revealed for some notable cases. The Musashi (MSI) family of RBPs are well known to promote stemness and pluripotency, including in embryonic and neural stem cells [13]. Recent studies found that MSI proteins form condensates in an IDR-dependent manner, with an extended poly-alanine tract contributing to self-associating interactions [14,15]. The exact role that MSI condensation plays in stem cell function is yet to be elucidated, however. Another recent example involves the function of YTHDF1 phase separation in spermatogonial stem cells. This RBP, containing multiple prion-like IDRs, forms cellular condensates that are essential for transdifferentiation into induced neural stem cell-like cells [16]. Molecular analysis suggested that YTHDF1 phase separation controls the expression of genes that are important for this process. With many RBPs yet to be explored specifically within the context of stem cell function, this area of study remains promising for linking condensate and cell function.

A second category of condensed complexes to be broadly characterized are those that do not involve RNA and typically involve important cell signaling complexes [17]. These condensates contain protein–protein complexes that, like their RNP counterparts, have some recurring molecular features. Most notably, multivalent interactions appear to be a driver of their formation, wherein the larger protein complexes become less soluble and increase the energetic favorability of condensates (Figure 1). Such multivalency can result from obligate oligomerization, such as that seen in condensates containing coiled-coil proteins that necessarily form higher-order complexes [18,19,20,21]. A more encompassing mechanism involves proteins containing repeating protein interaction domains, such as the ubiquitous PDZ domain [22,23,24]. Such arrays provide a modular platform for multiple partner interactions within a single protein. The condensation of these complexes is influenced by the number of modular domains (e.g., the stoichiometry of the bound complex), as well as by the length and specific sequence of the linkers between them [25,26]. In both RNP and multivalent protein complexes, the condensation process can be further regulated by additional factors, notably, post-translational modifications such as phosphorylation [27,28].

## 3. Phase Separation Drives the Formation of Biological Condensates Essential to Development

Although phase separation, as discussed herein, is a phenomenon described relatively recently, biologically important condensates had been known for many decades prior to their characterization. Particularly relevant to development are germ granules, such as P granules, which are RNP complexes that promote totipotency in germ cells and are essential for the development of egg and sperm [29]. The localization of these cytoplasmic condensates is polarized during asymmetric cell division, which is dependent on their phase-separated condensation, and influences the cell fate of the inheriting daughter cell through the regulation of co-segregating RNAs [30]. Germ granules redistribute in the cell as they increase in quantity and size and in some cases continually dissolve and condense to achieve their desired distribution within the cell [31]. Their ability to phase-separate is essential for their partitioning in the germ cell, and condensate disruption leads to defective early development. Initial descriptions of the liquid droplet-like qualities of P granules demonstrated that granule formation occurs through phase separation [31]. P granules are non-homogenous assemblies of core RGG-containing PGL proteins surrounded by the IDR-containing RBP MEG-3 [32,33]. This complex assembly is further regulated by phosphorylation [34]. Similar processes underlie the assembly of polar granules in *Drosophila* oocytes. Here, maternal mRNAs important for germ cell fate adoption phase-separate with the RNA helicase Vasa [35]. In contrast to P granules, proteins within polar granules are homogenously distributed; the self-association of mRNAs allows for their unique distribution within droplets [36,37]. These studies highlight the important nature of phase separation in germ granule assembly and regulation and the sophistication with which their cognate molecules are distributed.

Subnuclear organization has also long been appreciated to occur in the absence of formal membranes and, as for germ granules, is now understood to result from condensate activities [38]. Among numerous subnuclear structures, the nucleolus is the most prominent and developmentally relevant. Organized around rDNA clusters, the nucleolus serves as the site of ribosomal gene transcription and initial ribosome subunit assembly [39]. Ribosome biogenesis is essential for proper proteostasis, a critical requisite for maintaining stemness and suppressing stem cell aging [40]. The nucleolus was more recently identified as a prominent site of heterochromatin organization and as a regulator of gene expression, including in stem cells [40,41], the disruption of which can lead to defects in stem cell function and pluripotency [42,43]. Nucleolus-associated domains are organized heterochromatin regions that include significant amounts of the genome in embryonic stem cells and may coordinate with those regions associated with nuclear lamin to control gene expression [41]. Nucleolar condensation was first suggested by observations of its liquid-like behaviors in *Xenopus* oocytes, including fluid dynamics and spherically shaped droplets capable of fusion [44]. Nucleophosmin (NPM1), an RNA-binding protein, is among the first prominent nucleolar proteins subsequently characterized to phase-separate in vitro with dynamics similar to those of nucleoli themselves [45]. Amazingly, when combined with another nucleolar component, fibrillarin (FIB1), NPM1 and FIB1 co-separate, though into distinct internal phases within the condensed droplets, reminiscent of their respective subnucleolar compartments (the dense fibrillar center and the granular center, respectively) in vivo [46]. This multiphase partitioning appears dependent on both disordered protein regions as well as RNA-binding domains [46].

In neural stem cells, the nuclear protein Trnp1 was shown to interact with and regulate the nucleolar structure, along with other condensed subnuclear structures. This activity relies on its ability to phase-separate, which is afforded through a conserved N-terminal IDR. Interfering with Trnp1 condensation alters nucleolar structure and stem cell proliferation [47]. The disruption of nucleolar condensation has other notable implications. For example, the loss of RNA polymerase I (Pol I) and rRNA transcription in embryos or iPSCs results in a disrupted nucleolar condensate structure and a collapse into a singular nucleolus, a marker of stress, followed by cell death. The mitotic disruption of Pol I interferes with nucleolar reformation following cell division, leading to fragmented nucleolar condensates [48]. Furthermore, the rare genetic disorders brachyphalangy, polydactyly, and tibial aplasia syndrome were recently found to result from frameshift mutations in HMGB1, a conserved nucleosome regulator. These HMGB1 frameshifts alter the acidic property of the IDR tail, leading to impaired phase separation and aberrant nucleolar partitioning and dysfunction [49]. Taken together, these studies firmly establish biomolecular condensation as an essential aspect of subnuclear organization, particularly in nucleolar assembly and function.

## 4. Phase Separation in Chromatin Organization and Gene Expression

In addition to these long-appreciated subnuclear structures, more recent studies illuminated an immense level of genome organization within the nucleoplasm. Specifically, chromatin regions across the genome can assemble into highly ordered topologically associated domains (TADs), serving as a mode of 3D genome organization [50]. Connections within these domains, as well as among independent TADs, play critical roles in the regulation of gene expression during development [51]. TADs play a crucial role in stem cell function, including the control of pluripotency and differentiation [52]. Indeed, TAD disruption leads to aberrant genome regulation and development and impaired viability in a variety of model organisms [53]. Fundamental to TAD structure and function are specific chromatin configurations at loci across the genome that dictate TAD compartmentalization [50,54]. Although incompletely defined, a role for phase separation in TAD patterning, chromatin structure, and the assembly of transcriptional complexes was suggested to assist in gene expression control [55]. Examples of such behavior controlling stem cell function are detailed below.

A direct role of condensates in controlling gene expression was revealed at the level of transcription factors (TFs) bound to specific genomic loci (Figure 2). This was first hypothesized as a model to explain the dynamic kinetics and cooperative output of transcription [56]. Structurally, many TFs contain a globular DNA-binding domain (DBD) flanked by unstructured, intrinsically disordered activation domains (ADs) that together mediate binding at specific promoter and enhancer sites. Disordered AD sequences resemble those found in other protein condensates and are typified by an abundance of acidic residues [57,58]. Not surprisingly then, much evidence has now emerged spotlighting the capacity for condensation of TF complexes, which typically contain DNA and RNA along with co-regulatory proteins [59]. This phenomenon was first described to involve TF complexes associated at super-enhancers, genomic regions that assemble in a cooperative manner to ensure robust gene expression, notably including those involved in stem cell fate and identity [60,61]. These studies revealed that nuclear condensates containing the transcriptional coactivators BRD4 and MED1 form at super-enhancer sites. These proteins phase-separate in vitro in an IDR-dependent manner, with MED1 being competent to recruit BRD4 for co-segregation along with RNA Pol II [61]. Studies in embryonic stem cells revealed that MED1 condensates associate with TF-bound chromatin and with condensates containing Pol II in a transcription-dependent fashion [60]. Subsequent studies revealed this phenomenon to be more expansive, with diverse TFs using a disordered AD to phase-separate together with Mediator complexes [62]. The Kruppel-like factor 4 (KLF4) transcription factor is a critical component of stem cell reprogramming cocktails for the generation of iPSCs, cooperating with Oct4 to activate pluripotency genes such as NANOG. Condensation of KLF4 occurs with DNA sequences of the NANOG promoter and induces the recruitment of Oct4 complexes to promote iPSC reprograming [63]. The SWI/SNF chromatin remodeling factor SS18 was also recently found to phase-separate through its tyrosine-based IDR. Notably, SS18 condensation assembles the SWI/SNF complex during pluripotent-somatic transition, a critical stem cell function during development [64]. Together, these studies suggest that the condensation of chromatin regulatory and TF complexes plays key roles in gene expression and may differ between stem and somatic cells to control cell fate transitions during development.

Heterochromatin protein 1 (HP1) is a non-histone heterochromatin regulator essential for the epigenetic silencing of gene expression. Its effects occur through the compaction of chromatin across large genomic regions and the recruitment of additional regulators to the resulting complex [65]. HP1 is critical for ESC pluripotency and self-renewal in adult stem cells [66,67]. Two seminal studies recently identified phase separation as an important aspect of HP1-dependent chromatin regulation [68,69] (Figure 2). HP1-dependent phase separation is facilitated by DNA binding and the phosphorylation of its disordered N-terminus and leads to DNA compaction consistent with heterochromatin. Interestingly, droplets recruit repressive nucleosome factors while excluding transcription factors, which is also consistent with HP1-mediated gene silencing [68]. Whereas HP1 remains fluid and dynamic within droplets, compacted DNA has a constrained, immobile disposition [70]. Further studies suggested that HP1 oligomerization drives LLPS through conformational changes that expose protein sequences that facilitate multivalent nucleosome assemblies [71]. Studies observing the LLPS of additional chromatin complexes implicated nucleosome spacing and histone acetylation as additional regulatory mechanisms [72].

Finally, phase separation was described by some studies to impact the assembly and regulation of higher-order TAD genome assemblies in ESCs that impact cell fate decision making. Of particular interest is the Oct4 homeodomain transcription factor, which plays a central role in ESC self-renewal and in the maintenance of an undifferentiated state, as well as in the reprogramming of induced pluripotent stem cells [73] (Figure 2). Recent work identified a role for Oct4 condensation in controlling super-enhancer-associated TAD organization [74]. Specifically, TADs were shown to undergo reorganization during somatic cell reprogramming that was associated with changes in the transcriptional profile and in cell fate. These reorganizing changes were associated with Oct4 phase separation, while interfering with its condensation disrupted TAD dynamics. A disordered protein sequence was found to be both required and sufficient for Oct4 condensate formation [74]. The involvement of phase separation at various levels of genome organization and regulation highlights its critical role in cell fate adoption.

## 5. Phase Separation in Developmental Signal Transduction Pathways

Following the discovery of condensates in the nucleus, studies identifying the role of phase separation in classical signal transduction pathways quickly emerged [17]. Lacking the RNA-/DNA-binding properties of nuclear condensates, many of these classical signaling complexes were found to induce phase separation via multivalent interactions occurring through modular protein interaction domains. The ability of signaling complexes to phase-separate likely affords an increased efficiency in signal output while sequestering selective regulatory components [17]. Below, we will highlight recent examples of those signaling pathways most relevant to stem cell signaling during development.

Wnt signaling is an evolutionarily conserved pathway that controls many fundamental aspects of animal development. In the canonical pathway, binding to the frizzled receptor leads to the disassembly of the β-catenin ‘destruction complex’, formed by its assembly with axin, APC, GSK3β, and CK1α, thereby allowing for β-catenin nuclear translocation and the expression of Wnt target genes [75] (Figure 3). Studies found that the axin scaffold protein undergoes IDR-dependent condensation in vitro, which is competent to recruit β-catenin as well as GSK3β and CK1α [76]. Moreover, phase separation of this complex enhances β-catenin phosphorylation, controlling its stability and regulating Wnt signaling output. APC was further shown to regulate the size and liquid-like dynamics of axin-scaffolded condensates [76]. An independent study found that dishevelled-binding antagonist of β-catenin 1 (DACT1) also undergoes IDR-dependent condensation to control Wnt signaling. Cellular DACT1 condensates appear to be uniquely cytoplasmic, and authors suggested a model in which the sequestered CK2 within them may prevent its participation in Wnt signaling [77].

Another conserved and fundamental signaling pathway recently found to rely on phase separation for function is the Hippo kinase pathway. Canonically, Hippo and Warts kinases (Mst1/2 and LATS1/2 in mammals) coordinate to phosphorylate the transcriptional coactivator Yorkie (Yki; YAP/TAZ in mammals) to direct 14-3-3-mediated cytoplasmic retention. Upon Hippo deactivation by upstream components, nuclear translocation and Yki-dependent activation of cell growth and proliferation genes are promoted [78] (Figure 3). Two studies first identified the ability of the YAP/TAZ transcriptional regulators to phase-separate as a mechanism for driving the nuclear reorganization of the genome structure [79,80]. First, hyperosmotic stress was found to induce the nuclear condensation of YAP/TAZ along with the TEAD1 transcription factor at sites determined to be open, accessible chromatin and assembled as super-enhancer sites [79]. The condensates also recruited RNA polymerase II to initiate gene transcription. Furthermore, YAP constructs that could not undergo cytoplasmic phase separation were impaired in nuclear translocation. Second, the additional transcriptional regulators BRD4, MED1, and CDK9 were found to be recruited to TAZ condensates, which were disrupted by Hippo activity [80]. Domain mapping implicated the TAZ coiled-coil and WW domains in condensate formation (Figure 3), suggesting that they serve as sites of self-association to establish multivalent assemblies competent for condensate formation. Subsequently, upstream Hippo regulators were also identified to phase-separate to control signaling [81]. Hippo regulation is complex, and authors found that several distinct complexes form condensates that can coalesce to form multiphase structures that ultimately activate or suppress Hippo signaling. These results were confirmed in an independent study that also identified IDRs within Hippo regulatory proteins that facilitate their condensation [82]. Notably, studies in ESCs found that the regulation of YAP1 compartmentalization within kinase condensates can control YAP1 function in endoderm differentiation and cell fate decision making [83].

Lastly, components of the fibroblast growth factor (FGF) signaling pathway were recently shown to phase-separate. FGF signaling occurs through a receptor tyrosine kinase that activates typical Ras, PI3K, and PLCγ intracellular signaling and can be modulated by tissue-specific heparin sulfate [84]. FGF signaling plays an important role in promoting stem cell self-renewal and pluripotency maintenance, while also preventing quiescence [85]. Initial studies revealed that the FGF ligand can phase-separate together with heparin sulfate, mimicking a crowded environment at the cell surface, suggesting that condensation facilitates its interaction with the receptor to initiate signaling [86]. Other concurrent studies found that activated, phosphorylated FGF receptors (FGFRs) form multivalent effector complexes that undergo phase separation [87]. These FGFR-dependent condensates recruit PLCγ and activate its enzymatic function. Interestingly, the Ets-2 responsive factor ERF, a nuclear effector of the FGF pathway, can also phase-separate in a manner sensitive to phosphorylation from FGFR signaling [88]. Phase separation is an important regulator of nuclear ERF localization and function, which is notable, since ERF plays a critical role in transcriptional activation during early zygotic development [88,89]. Thus, phase separation appears to control the developmentally important FGF signaling pathway at several levels.

## 6. Phase Separation in Asymmetric Cell Division

Asymmetric cell division (ACD) is a conserved mode of stem cell division that produces non-identical daughter cells essential for the generation of cell diversity throughout development [90,91]. ACD requires two critical processes, i.e., cell polarity and mitotic spindle orientation, to be coupled in a manner that leads to the unequal segregation of cell fate determinants. Typically, one cell inherits self-renewal factors, whereas the other contains those necessary for differentiation. Although the core ACD components have been known for decades, a role for LLPS in their cortical assembly and function was only recently revealed. The studies described below utilized *Drosophila* neural stem cells (neuroblasts, NBs) as a stem cell model system (Figure 4), although the protein complexes involved are highly conserved, as are their general functions in many asymmetrically dividing cell types across taxa. NBs polarize along an apical–basal axis and orient their spindles to produce ACD, which generates a self-renewing NB and a neural precursor cell [92].

The first described ACD protein complex involved in phase separation is the basal Numb complex. Numb is an antagonist of Notch signaling, and its partitioning into the basal daughter cell is an important negative regulator of stemness (Figure 4). In NBs, Numb segregates to the basal cortex in complex with its adaptor protein partner of numb (Pon), the disruption of which leads to defective differentiation and the expansion of the stem cell pool [93]. The Numb/Pon complex was found to phase-separate in vitro, using multivalent assembly wherein Numb interacts with tandem Pon-binding motifs [94]. While neither component is competent to phase-separate alone, the Numb/Pon complex readily forms dynamic condensates. The structural resolution of this multimeric complex provided a molecular basis for its high-affinity assembly that appears necessary for phase transition. Notably, mutations that disrupt Numb/Pon LLPS impair its basal polarization, leading to altered Notch signaling following NB cell division.

Following shortly after the discovery of Numb complex phase separation, a similar process for the evolutionarily conserved Par complex was identified (Figure 4). This complex consists of the Par3/Par6 scaffold proteins bound together with the atypical protein kinase C (aPKC) enzyme [95]. The Par complex is apically localized in NBs, and as in diverse other cell types, aPKC-mediated substrate phosphorylation is the essential, underlying mechanism for generating apical–basal cell polarity. In this case, Par3 can phase-separate in isolation due to its intrinsic homomeric oligomerization capacity [96]. This feature in turn allows Par3 to recruit both Par6 and aPKC into condensates. Active aPKC phosphorylates Par3, leading to the dispersion of droplets, suggesting that condensation could serve as a regulatory mechanism for the output of the polarity complex. Here again, the disruption of Par3/Par6 phase separation impairs NB polarity and provokes defects in asymmetric cell fate acquisition [96]. In radial glial cells, which are mammalian neural stem cells, Par3 phase-separates into an alternative complex containing Numb to control the balance between proliferation and differentiation during development [97].

More recently, we described a role for phase separation in the function of the apical spindle positioning complex [19] (Figure 4). The core complex consists of partner of inscuteable (Pins) bound to mushroom body defect (Mud) and discs-large (Dlg), each of which interact with microtubule motor proteins to direct spindle forces. Initially postulated to assemble with the polarity complex through the inscuteable (Insc) scaffold protein, Insc was later found to compete against the formation of the Pins/Mud complex, leaving an unresolved paradox for how polarity and spindle orientation complexes exist mutually at the apical cortex [98,99,100]. We found that Pins undergoes phase separation when bound to Mud but not when bound to Insc, suggesting that condensate formation may contribute to selective complex assembly within the apical domain [19]. Phase separation was dependent on a Mud coiled-coil domain, consistent with multivalent interactions as a contributing force. Our studies also identified the actin-binding adducin protein, Hu li tai shao (Hts), as a Mud-binding protein that facilitates condensation through its C-terminal IDR and functions as an apical spindle positioning factor. The known role of adducin proteins in actin regulation suggests a potential role for the cytoskeletal cortex in the phase separation process as well. Overall, these studies suggest that phase separation plays a critical function in controlling both polarity and spindle orientation complex assemblies during ACD in stem cells.

## 7. Phase Separation in Disease

As detailed above, many fundamental cell processes rely on phase separation during development as well as throughout adulthood for proper function and output. As such, alterations in condensate function could result in aberrant signaling or deviation from normal homeostasis, resulting in disease. For example, some disease-associated mutations were shown to abnormally trigger condensates, while others cause protein aggregation rather than fluid droplet formation. This has been prominently observed in amyotrophic lateral sclerosis (ALS) and other neurodegenerative diseases. The ALS-associated genes FUS and TDP-43, among other RBPs, undergo phase separation, with liquid-like droplets converting to an aggregated state over time. ALS patient mutations hasten this transition to a more solid-like state [101,102]. Mutations in the C-terminus of uniquilin-2 (UBQLN2), a component of the proteosome system responsible for protein degradation, were shown to cause X-linked ALS (~10% of the cases are familial). Mutations in the proline-rich region of UBQLN2, Pxx, promote oligomerization and condensation [103]. This impairs downstream functions including protein degradation, which causes the accumulation of peptides and protein aggregates [104]. Similarly, the loss of the N-terminal IDR of RB1CC1, an important autophagy component, results in impaired formation of the autophagosome via phase separation [105]. The impacts of defective autophagy have been widely reviewed [106] and include debris accumulation, cellular aging, and degeneration [107]. The enzyme superoxide dismutase (SOD1) can also harbor mutations that induce aggregates and misfolded proteins that are toxic to nerve cells [108].

Mutations that result in proteins with altered chemical properties can also impair or promote atypical condensate assembly. SHP2 (PTPN11), an allosteric enzyme involved in RAS–mitogen-activated protein kinase (MAPK) activity, is responsible for Noonan syndrome and is commonly detected in juvenile myelomonocytic leukemias, a condition characterized by errors in blood cell development. Normal signaling involves autoinhibition of SHP2 via intramolecular interactions between the catalytic domain and the NH2 terminus. Mutations in SHP2 prevent this interaction and favor an active conformation that stimulates an abnormal LLPS, which promotes RAS–MAPK activation [109], leading to the hyperproliferation of blood cells [110].

Abnormal phase separation can also be provoked by small nucleolar RNA (lncRNA). LncRNAs are well-known moderators of intracellular signaling pathways involved in important regulatory functions including cell growth and differentiation [111]. Moreover, numerous studies demonstrated the importance of RNA in phase separation, and recent studies also suggested a role in disease. The tumor suppressor pathway Hippo relies on several upstream factors to mediate signaling, and previous studies identified lncRNAs as important monitors of this pathway [112,113]. Attempts to decipher the specific role of lncRNAs revealed that some lncRNAs can support abnormal condensation. The lncRNA small nucleolar RNA host gene 9 (SNHG9) was shown to inhibit the activity of the Hippo core kinase LATS1, via induction of phase separation [114]. While LATS1 can undergo phase separation due to the presence of the prion-like domain (PLD), SNHG9 potentiates LLPS. Concomitant with these findings, increased levels of SNHG9 are often observed in many breast cancers, and this is accompanied by increased Yes-associated protein (YAP) activity.

Phase separation also alters the activity of pathways regulating intracellular processes. As discussed in this review, condensates can fuse or separate to sequester factors necessary for signaling. With respect to the Hippo pathway, inclusion of the paraspeckle protein non-POU domain-containing octamer-binding protein (NONO) promotes the phase separation of TAZ and [115] encourages tumorigenic growth, while decreased levels are consistent with a low transcription of YAP target genes. Elevated levels of NONO are common in hyperplasia and are correlated with poor prognosis in many cancers, including melanoma, breast cancer, and neuroblastoma [116,117]. Furthermore, recent studies demonstrated that YAP phase separation is facilitated by its transcriptional co-activator SRC-1, and loss of SRC-1 decreases the expression of YAP target genes through impaired condensate formation [118]. Thus, NONO- and SRC-1-mediated phase separation provides another focus for pharmacological inhibition. The importance of these findings is twofold. First, they advance the knowledge of how growth-promoting signaling is compartmentalized, and second, they provide new research avenues for the discovery of potential drug targets. Numerous other cancer-associated pathways have also been shown to involve LLPS and may play a role in the pathogenesis of certain tumors [119,120].

## Figures and Tables

**Figure 1 jdb-11-00045-f001:**
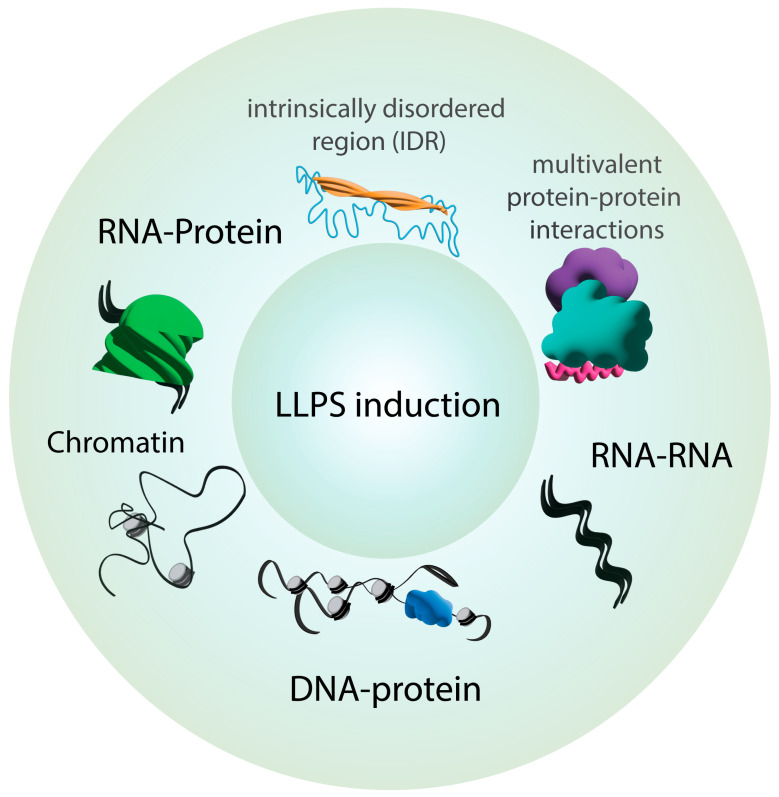
Molecular features commonly found in biomolecular condensate components. Some chemical properties of proteins can drive LLPS induction. Intrinsically disordered regions (IDRs), multivalent protein–protein interactions, and nucleic acids are commonly found in membranelle compartments and are associated with the formation of biomolecular condensates.

**Figure 2 jdb-11-00045-f002:**
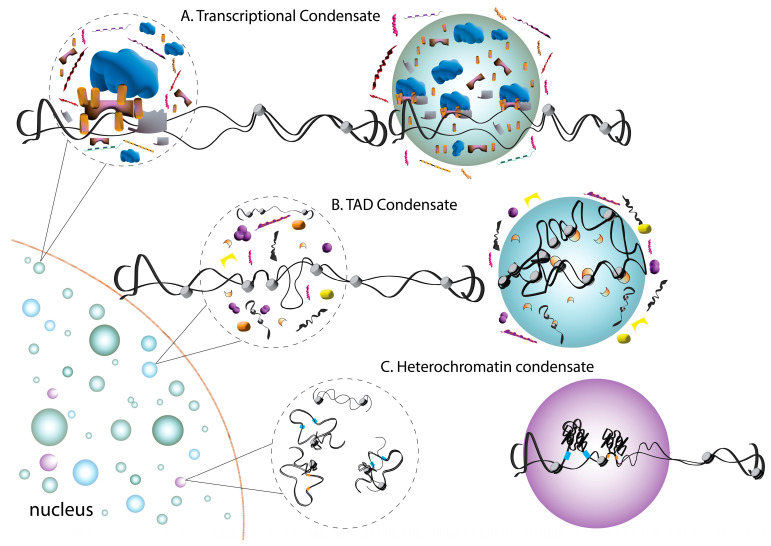
Summary of phase separation events in the regulation of gene expression. Changes in gene expression occur in condensates that sequester the transcriptional machinery. (**A**) Homogenous mixtures (dashed circles) contain accessory proteins (purple, pink, orange) and co-activators (gold, gray) that facilitate transcription (DNA; black, histones; grey) in response to a stimulus or return to the homogenous state in the absence or with the withdrawal of stimuli. Transcriptional condensates often form through IDRs found within the activation domain of specific transcription factors. (**B**) Topologically associated domains (TADs) are three-dimensional organizations of chromatin regions that span large genomic distances to control gene expression and contain histone proteins and marks (gray and orange). Chromatin loops are often conserved across cell types and are monitored by regulatory elements (yellow, pink). (**C**) Gene repression occurs via condensate-mediated sequestration of heterochromatin protein 1 (HP1) (blue) into heterochromatin condensates. These condensates form around promoters on chromatin (black) and histones (gray). Histone-associated HP1 (orange) promotes gene silencing and the organization of higher-order chromatin structures.

**Figure 3 jdb-11-00045-f003:**
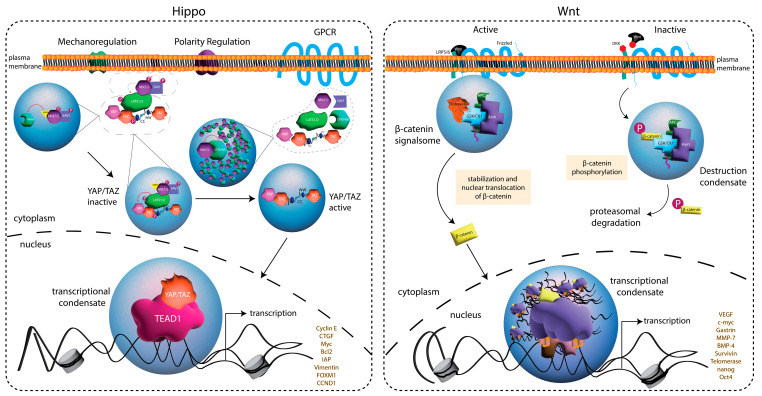
Summary of the signaling pathways regulated by phase separation during development. (**Left**): Hippo signaling pathway regulates cell growth, proliferation, and survival, including in many stem cells. Mechanoregulation, polarity complexes, and GPCR signaling are upstream regulatory signals modulating Hippo activity. Hippo (Mst1/2) signaling is a phosphorylation cascade that inactivates STRIPAK. This leads to the phosphorylation of YAP/TAZ by LATS1/2, thereby preventing the nuclear translocation of the transcriptional regulator. Partitioning of Mst1/2 and STRIPAK to distinct condensates prevents YAP/TAZ phosphorylation, leading to the nuclear translocation and transcriptional activation of target genes. (**Right**)*:* Wnt signaling also plays a critical role in development and occurs through binding to the frizzled receptor and LRP5/6 co-receptor (‘Active’). Wnt pathway activation is facilitated by the formation of β-catenin signalosome condensates consisting of disheveled, GSK/CK1 kinases, APC, and axin. This complex causes the stabilization of β-catenin by preventing its phosphorylation and allowing its nuclear translocation for the activation of target genes. The absence of Wnt (‘Inactive’) promotes the phosphorylation of β-catenin within distinct condensates containing a destruction complex composed of active GSK/CK1 kinases, APC, axin, and β-catenin. Phosphorylated β-catenin is targeted for proteasomal degradation.

**Figure 4 jdb-11-00045-f004:**
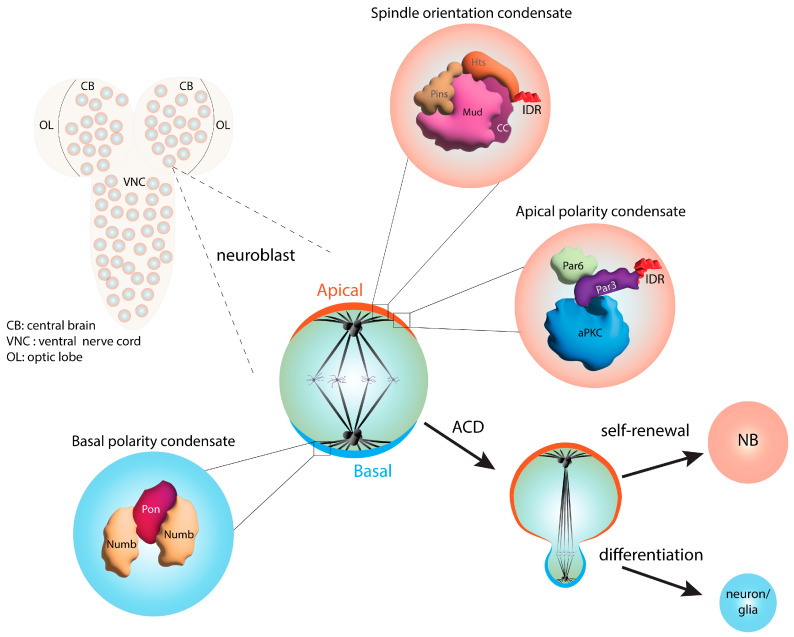
Summary of phase separation functions in asymmetric cell division. *Drosophila* neuroblasts (NBs), like most stem cells, divide via asymmetric cell division (ACD). ACD is established by the coordination of mitotic spindle orientation and cell polarity cues. Distinct condensates containing apical (orange) and basal (blue) complexes have been identified. The apical polarity complex (orange) consists of the conserved proteins atypical protein kinase C (aPKC), Par3, and Par6. The intrinsically disordered region of Par3 facilitates the phase separation of this complex. The apical spindle orientation complex contains Pins, Mud, and Hu li tai shao (Hts). Formation of the spindle orientation condensate is promoted by multivalent protein–protein interactions mediated by the Mud coiled-coil domain and an IDR within the Hts C-terminus. The basal domain (blue) of NBs contains the differentiation factors Numb and partner of numb (Pon). These components form a basal polarity condensate through multivalent interactions of Numb with tandem binding sequences in Pon. The phase-separated complexes are inherited by the daughter cell that will differentiate to form neurons or glial cells, following ACD.

## Data Availability

No new data were created or analyzed in this study. Data sharing is not applicable to this article.

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
