# Peer review of "Phase Separation as a Driver of Stem Cell Organization and Function during Development"

_jdb, 2023, doi:10.3390/jdb11040045_

Round 1

Reviewer 1 Report

Comments and Suggestions for Authors

This Review by Amalia Parra and Christopher Johnston seeks to summarize the field of biomolecular condensation and its potential role in stem cell organization and function. They first provide (1) an introduction and (2) a summary of phase separation as a process to create subcellular compartments, then put forth evidence of a role for phase separation in four cellular functions important in stem cells, (2) nucleolar form and function, (3) chromatin organization and gene expression, (4) signal transduction, and (5) asymmetric cell division. Overall, the authors’ summary of the field of biomolecular condensation and many studies connecting it to these cellular functions is comprehensive and solid, but I am left feeling as though this is a review about roles for phase separation in general cell biology, which has been largely covered elsewhere, rather than specifically for stem cell biology, which would be a novel concept. I suggest the authors edit each section, especially the nuclear organization sections, focusing on streamlining the generic condensation information and expand on definite or potential connections to stem cell function.

For example, in section ‘2. Phase separation as a dynamic mode of cellular organization’, the authors mention that IDRs and RBPs are commonly associated with condensation, but do stem cells express different RBP or IDR-containing protein content than other cell types? Do they have more or different condensates in cytoplasm or nucleoplasm than other cell types? 

Or, in the section about nucleolus, is there any known specific link between the condensate nature of the nucleolus and its stem-cell related functions (proteostasis, suppression of stem cell aging)? They can expand on the point mentioned in lines 150-152 that “Disruption of phase separation leads to altered nucleolar structure and has recently been suggested to underlie two-cell developmental arrest as well as a myriad of genetic disorders [40, 41].“ 

Again, in section 4, how might transcriptional condensates and heterochromatic phase separation differ between stem cells and other cell types? What specific nuclear organization exists in stem cells that is influenced by condensation? 

The signaling and asymmetric division sections are more clearly stem-cell relevant.

Minor text comments: 

I urge the authors to be careful with their wording– phase separation, phase transition, condensation, Liquid-liquid phase separation all refer to slightly different physical phenomena. Especially with uses of LLPS, which refers specifically to separation between two liquid-like phases. The field is coalescing on ‘condensation’ as the most generic term, I suggest replacing relevant instances of LLPS with condensation.

In lines 86-87, the authors mention condensates that have ‘membrane delimited localization’ which makes it sound like they are surrounded by a membrane but in fact they are referring to condensates that form in the cytoplasm but localize to the inner side of the plasma membrane. I suggest rewording this sentence for clarity.

A minor stylistic comment: it is strange to call out certain labs by name in the text while having most references as numbers. I suggest rewording to not include specific lab or author names in the main text. 

Figure comments:

Overall I don’t find the figures very illuminating, and in my opinion they have the same lack of focus on stem cell biology as the main text. It would be great to see some stem cell-specific functions mediated by condensation illustrated in the figures. 

Figure 1 specific comments: Some of the things listed in the “LLPS induction” circle are interactions (RNA-protein) and others are just one object (IDR), I suggest renaming to something like ‘interactions that drive condensation’ and then could list ‘IDR-IDR, RNA-RNA, RNA-protein …’. The 3D representation of the fusion, partitioning and fission behaviors are a bit strange, what is the significance of the central third lobe? And after fission why do the condensates still have a delineation? Similar to the comment above, I’d appreciate a visual representation of how material properties of condensates or partitioning of certain factors into/out of those condensates would contribute to stem cell function. 

Fig 2 specific comments: I don’t understand the difference between schematics before and after stimulus, or what is trying to be communicated about the role of condensation in these processes. Are the authors trying to say each of these condensates regulated by a specific stimulus? What would be the stimulus? I don’t think the field agrees that TADs are formed by condensates. 

Fig 3. Is every step in these pathways regulated by condensation? 

Fig 4 is more interpretable

Author Response

We thank the reviewer for their thorough read of the manuscript and thoughtful suggestions for improvement. We have attempted to make the majority of these changes and hope the reviewer agrees that the revised manuscript has improved for acceptance. Below are the individual comments along with responses to each:

  • Overall, the authors’ summary of the field of biomolecular condensation and many studies connecting it to these cellular functions is comprehensive and solid, but I am left feeling as though this is a review about roles for phase separation in general cell biology, which has been largely covered elsewhere, rather than specifically for stem cell biology, which would be a novel concept.
    • Although we agree that our review does not solely focus on stem cell-specific discussions, we feel it is important to provide readers with general background of some important features of condensate biology. We hope this will make for a more comprehensive review for the reader. As detailed for the reviewer’s specific comments that follow, we have added a few additional stem cell examples, but we do not wish to remove what we feel is important contextual information that may nevertheless read as more general cell biology.
  • For example, in section ‘2. Phase separation as a dynamic mode of cellular organization’, the authors mention that IDRs and RBPs are commonly associated with condensation, but do stem cells express different RBP or IDR-containing protein content than other cell types? Do they have more or different condensates in cytoplasm or nucleoplasm than other cell types? 
    • Section 2 was originally intended to serve as a general overview and introductory extension. To add stem cell context, we have added references to the importance of RBPs in stem cell function and a short discussion of a couple specific examples that are known to phase separate according to the general properties discussed in this section [Lines 90-103]. We have also moved the small paragraph about the intent and goal of the review to the end of section 1 rather than the end of section 2 to further reflect these changes.
  • Or, in the section about nucleolus, is there any known specific link between the condensate nature of the nucleolus and its stem-cell related functions (proteostasis, suppression of stem cell aging)? They can expand on the point mentioned in lines 150-152 that “Disruption of phase separation leads to altered nucleolar structure and has recently been suggested to underlie two-cell developmental arrest as well as a myriad of genetic disorders [40, 41].“
    • Studies specifically examining the condensate nature of the nucleolus to aging were not evident. We have, however, added additional discussion points regarding developmental arrest and genetic disorders, including specific genes involved and the relationship to condensate function.
    • We also added an additional reference to a phase separating regulator of nucleolar function in neural stem cells [Lines 167-171].
  • Again, in section 4, how might transcriptional condensates and heterochromatic phase separation differ between stem cells and other cell types? What specific nuclear organization exists in stem cells that is influenced by condensation?
    • We have added discussion and references to two additional studies investigating the role of condensate formation of stem cell reprogramming complexes [Lines 219-226] for additional context regarding how such processes may differ in stem cells compared to differentiated cells.
  • Additional references have been added to other sections that provide links between condensate formation and control of various stem cell functions:
    • Regulation of YAP1 condensates in ESC fate decisions [Lines 304-306].

Minor comments:

  • I urge the authors to be careful with their wording– phase separation, phase transition, condensation, Liquid-liquid phase separation all refer to slightly different physical phenomena. Especially with uses of LLPS, which refers specifically to separation between two liquid-like phases. The field is coalescing on ‘condensation’ as the most generic term, I suggest replacing relevant instances of LLPS with condensation.
    • We have attempted to avoid confusing terminology by using “condensates/condensation” throughout the manuscript as a broader term.
  • In lines 86-87, the authors mention condensates that have ‘membrane delimited localization’ which makes it sound like they are surrounded by a membrane but in fact they are referring to condensates that form in the cytoplasm but localize to the inner side of the plasma membrane. I suggest rewording this sentence for clarity.
    • We have clarified this sentence to more simply state “…and typically involve important cell signaling complexes.”
  • A minor stylistic comment: it is strange to call out certain labs by name in the text while having most references as numbers. I suggest rewording to not include specific lab or author names in the main text. 
    • We have removed the instances of specific lab mentions by name for references.

Figure suggested changes:

  • Figure 1 specific comments: Some of the things listed in the “LLPS induction” circle are interactions (RNA-protein) and others are just one object (IDR), I suggest renaming to something like ‘interactions that drive condensation’ and then could list ‘IDR-IDR, RNA-RNA, RNA-protein …’. The 3D representation of the fusion, partitioning and fission behaviors are a bit strange, what is the significance of the central third lobe? And after fission why do the condensates still have a delineation? Similar to the comment above, I’d appreciate a visual representation of how material properties of condensates or partitioning of certain factors into/out of those condensates would contribute to stem cell function. 
    • We agree that the representation of some condensate behaviors (e.g. fission/fusion) in the original figure did not convey those events well. The partitioning function was meant to depict what may be a more niche aspect of some condensates rather than a general aspect representative of stem cell function. Therefore, for clarity, we have decided to remove those features of the figure and instead make this figure a more simplified illustration of some common molecular aspects of condensate components. These are not intended to be stem cell specific, although those described in stem cells share many of these features.

  • Fig 2 specific comments: I don’t understand the difference between schematics before and after stimulus, or what is trying to be communicated about the role of condensation in these processes. Are the authors trying to say each of these condensates regulated by a specific stimulus? What would be the stimulus? I don’t think the field agrees that TADs are formed by condensates.
    • The “stimulus” labels were intended to generally convey that condensate formation can be a regulated process. Because this presents potential confusion within the figure, we have decided to remove the generic “stimulus” labels. Whether there is agreement that TADs form by condensates we honestly do not know. We have chosen to keep this as part of the figure simply as a reference to the studies described in the text that show connections between TAD components and condensate formation.

  • Fig 3. Is every step in these pathways regulated by condensation? 
    • Yes, although each condensate complex has been described in separate studies.

Reviewer 2 Report

Comments and Suggestions for Authors

In the manuscript entitled “Phase separation as a driver of stem cell organization and function during development” Authors review recent findings in liquid-liquid phase separation during polarized cell division that plays an important role in stem cells functioning during development. The manuscript is well written and contains illustrative material. I would recommend it for publication with some minor changes. Line 56 “liquid-liquid phase separation” should be changed on LLPS. Line 137 “Nucleolar-associated domains (NADs) are”. I did not find anywhere in the text NAds, thus the abbreviation could be excluded. The same is for Line 372 “ (NS) (JMMLs). Line 160 “topographically associated domains (TADs)” should be changed to topologically associated domains. Line 196, 209, 211 “eSC” should be changed to ESC.

Author Response

We thank the reviewer for their time and effort along with the helpful suggestions for improvement. We have now made all of these changes and hope the reviewer agrees that the revised manuscript has improved for acceptance. Below are the individual comments along with responses to each:

  • Line 56 “liquid-liquid phase separation” should be changed on LLPS.
    • We have made the suggested change to “LLPS”. Also note that in response to Reviewer 1, we have removed most references to "LLPS" in favor of the more general term "condensate" (and related derivations).
  • Line 137 “Nucleolar-associated domains (NADs) are”. I did not find anywhere in the text NAds, thus the abbreviation could be excluded.
    • We have removed the “NADs” acronym as suggested on line 137.
  • The same is for Line 372 “ (NS) (JMMLs).
    • We have removed the “NS” and “JMMLs” acronyms as suggested on line 372.
  • Line 160 “topographically associated domains (TADs)” should be changed to topologically associated domains.
    • We have changed “topographically” to “topologically” on line 160.
  • Line 196, 209, 211 “eSC” should be changed to ESC.
    • We have changed the instances of “eSC” to “ESC” as suggested.